# Statistical-Computational Trade-offs for Density Estimation

**Anders Aamand**
University of Copenhagen
aamand@mit.edu

**Alexandr Andoni**
Columbia University
andoni@cs.columbia.edu

**Justin Y. Chen**
MIT
justc@mit.edu

**Piotr Indyk**
MIT
indyk@mit.edu

**Shyam Narayanan**
Citadel Securities[*]
shyam.s.narayanan@gmail.com

**Sandeep Silwal**
UW-Madison
silwal@cs.wisc.edu

**Haike Xu**
MIT
haikexu@mit.edu

## Abstract

We study the density estimation problem defined as follows: given $k$ distributions $p_1, \ldots, p_k$ over a discrete domain $[n]$, as well as a collection of samples chosen from a "query" distribution $q$ over $[n]$, output $p_i$ that is "close" to $q$. Recently [1] gave the first and only known result that achieves sublinear bounds in *both* the sampling complexity and the query time while preserving polynomial data structure space. However, their improvement over linear samples and time is only by subpolynomial factors.

Our main result is a lower bound showing that, for a broad class of data structures, their bounds cannot be significantly improved. In particular, if an algorithm uses $O(n/\log^c k)$ samples for some constant $c > 0$ and polynomial space, then the query time of the data structure must be at least $k^{1-O(1)/\log\log k}$, i.e., close to linear in the number of distributions $k$. This is a novel *statistical-computational* trade-off for density estimation, demonstrating that any data structure must use close to a linear number of samples or take close to linear query time. The lower bound holds even in the realizable case where $q = p_i$ for some $i$, and when the distributions are flat (specifically, all distributions are uniform over half of the domain $[n]$). We also give a simple data structure for our lower bound instance with asymptotically matching upper bounds. Experiments show that the data structure is quite efficient in practice.

## 1 Introduction

The general density estimation problem is defined as follows: given $k$ distributions $p_1, \ldots, p_k$ over a domain $[n]^2$, build a data structure which when queried with a collection of samples chosen from a "query" distribution $q$ over $[n]$, outputs $p_i$ that is "close" to $q$. An ideal data structure reports the desired $p_i$ quickly given the samples (i.e., has fast query time), uses few samples from $q$ (i.e., has low sampling complexity) and uses little space.

---

[*]Work done as a student at MIT

[2]In this paper we focus on finite domains.

38th Conference on Neural Information Processing Systems (NeurIPS 2024).

In the realizable case, we know that $q$ is equal to one of the distributions $p_j$, $1 \leq j \leq k$, and the goal is to identify a (potentially different) $p_i$ such that $\|q - p_i\|_1 \leq \epsilon$ for an error parameter $\epsilon > 0$. In the more general agnostic case, $q$ is arbitrary and the goal is to report $p_i$ such that

$$\|q - p_i\|_1 \leq C \cdot \min_j \|q - p_j\|_1 + \epsilon$$

for some constant $C > 1$ and error parameter $\epsilon > 0$. The problem is essentially that of non-parametric learning of a distribution $q \in \mathcal{F}$, where the family $\mathcal{F} = \{p_1, \ldots p_k\}$ has no structure whatsoever. Its statistical complexity was understood already in [17], and the more modern focus has been on the structured case (when $\mathcal{F}$ has additional properties). Surprisingly, its computational complexity is still not fully understood.

Due to its generality, density estimation is a fundamental problem with myriad applications in statistics and learning distributions. For example, the framework provides essentially the best possible sampling bounds for mixtures of Gaussians [10, 18, 12]. The framework has also been studied in private [8, 7, 13, 15] and low-space [4] settings.

| Samples | Query time | Space | Comment | Reference |
|---|---|---|---|---|
| $\log k$ | $k^2 \log k$ | $kn$ | | [17, 11] |
| $\log k$ | $k \log k$ | $kn$ | | [2] |
| $\log k$ | $k$ | $kn$ | | [1] |
| $\log k$ | $\log k$ | $n^{O(\log k/\epsilon^2)}$ | precompute all possible samples | folklore |
| $n$ | $nk^\rho$ | $kn + k^{1+\rho}$ | any constant $\rho > 0$ | [16]+[14] |
| $\frac{n}{\log(k)^{1/4}}$ | $n + k^{1 - \frac{1}{\log(k)^{1/4}}}$ | $k^2 n$ | | [1] |
| $n/s$ | $k^{1-O(\rho_u)/\log s}$ | $k^{1+\rho_u}$ | *lower bound for any $\rho_u > 0$, sufficiently large $s$* | this paper |
| $n/s$ | $k^{1-\Omega(\rho_u)/\log s}$ | $kn + k^{1+\rho_u}$ | algorithm for half-uniform distributions | this paper |

Table 1: Prior work and our results. For simplicity the results stated only for the realizable case, constant $\epsilon > 0$, and with $O(\cdot)$ factors suppressed. The bound of [1] (row 6 of the table) is stated as in Theorem 3.1 of that paper. However, by adjusting the free parameters, their algorithm can be easily generalized to use $n/s$ samples for $n/s > n/\text{polylog}(n)$, resulting in a query time of $O(n + k^{1-\Omega(\epsilon^2)/s})$. Note that the term $1 - 1/s$ in their bound results in a larger exponent than $1 - 1/\log s$ in our upper bound. Furthermore, our algorithm is correct as long as $n/s \gg \log k/\varepsilon^2$ which is the information theoretic lower bound.

Table 1 summarizes known results as well as our work. As seen in the table, the data structures are subject to statistical-computational trade-offs. On one hand, if the query time is not a concern, logarithmic in $k$ samples are sufficient [17, 11]. On the other hand, if the sampling complexity is not an issue, then one can use the algorithm of [16] to learn the distribution $\hat{q}$ such that $\|\hat{q} - q\|_1 \leq \epsilon/2$, and then deploy standard approximate nearest neighbor search algorithms with sublinear query time, e.g., from [14]. Unfortunately both of these extremes require either linear (in $n$) sample complexity, or linear (in $k$) query time time. Thus, achieving the best performance with respect to one metric resulted in the worst possible performance on the other metric.

The first and only known result that obtained non-trivial improvements to *both* the sampling complexity and the query time is due to a recent work of [1]. Their improvements, however, were quite subtle: the sample complexity was improved by a sub-logarithmic factor, while the query time was improved by a sub-polynomial factor of $k^{1/(\log k)^{1/4}} = 2^{\log(k)^{3/4}}$.

This result raises the question of whether further improvements are possible. In particular, [1] asks: *To what extent can our upper bounds of query and sample complexity be improved? What are the computational-statistical tradeoffs between the sample complexity and query time?* These are the questions that we address in this paper.

- **Lower bound:** We give the **first** limit to the best possible tradeoff between the query time and sampling complexity, demonstrating a novel statistical-computational tradeoff for a fundamental problem. To our knowledge, this is the first statistical-computational trade-off for a *data structures* problem–if we allow for superpolynomial space, logarithmic sampling and query complexity

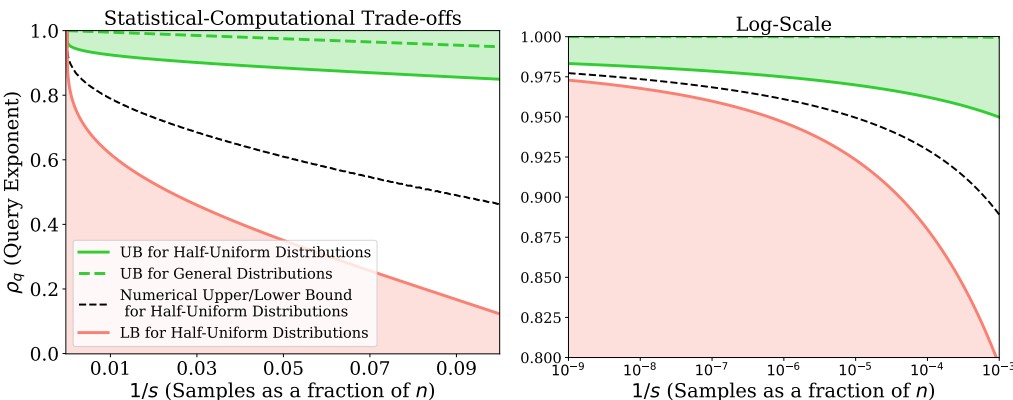

Figure 1: *Left:* Trade-off between $1/s$ (samples as a fraction of $n$) and the query time exponent $\rho_q$ for our algorithm for half-uniform distributions (solid green curve), the algorithm by [1] for general distributions (dashed green curve), our analytic lower bound (solid red curve), and a numerical evaluation of the bound from Theorem 3.2 (dashed black curve). We have fixed the space parameter $\rho_u = 1/2$. The plots illustrate the asymptotic behaviour proven in Theorem 3.1 and Theorem 4.2 that as $s \to \infty$, $\rho_q = 1 - \Theta(1/\log s)$ both in the lower bound and for our algorithm for half-uniform distributions. *Right:* The same plot zoomed in to the upper left corner with $1/s$ on log-scale.

is possible. As in [5, 6, 9, 3], we focus on data structures that operate in the so-called *list-of-points model*[3], which captures all bounds depicted in Table 1. Suppose that the query time of the data structure is of the form $k^{\rho_q}$. We show that, if the data structure is allowed polynomial space $kn + k^{1+\rho_u}$ and uses $n/s$ samples, then the query time exponent $\rho_q$ must be at least $1 - O(\rho_u)/\log s$. Therefore, if we set $s = \log^c k$ for some $c > 0$, as in [1], then the query time of the data structure must be at at least $k^{1-O(1)/\log\log k}$. That is, the query time can be improved over linear time in $k$ by at most a factor of $k^{O(1)/\log\log k} = 2^{O(\log k/\log\log k)}$. This shows that it is indeed not possible to improve the linear query time by a polynomial factor while keeping the sampling complexity below $n/\log^c n$, for any constant $c > 0$.

Our lower bound instance falls within the realizable case and in the restricted setting where the data and query distributions are "half-uniform", i.e. each distribution is uniform over half of the universe $[n]$. Note that showing a lower bound under these restrictions automatically extends to the agnostic case with general distributions, as the former is a special case of the latter.

Our construction takes lower bounds by [3] for set similarity search as a starting point. We adapt their lower bound to our setting in which queries are formed via samples from a distribution. The resulting lower bound is expressed via a complicated optimization problem and does not yield a closed-form statistical-computational trade-off. One of our technical contributions is solve this optimization problem for a regime of interest to get our explicit lower bound.

- **Upper bound:** We complement the lower bound by demonstrating a data structure for our hard instances (in the realizable case with half-uniform distributions) achieving sampling and query time bounds asymptotically matching those of the lower bound. We note that the existence of such an algorithm essentially follows from [3]. However, the algorithms presented there are quite complex. In contrast, our algorithm can be viewed as a "bare-bones" version of their approach, and as a result it is simple and easy to implement. To demonstrate the last point, we present an empirical evaluation of the algorithm on the synthetic data set from [1], and compare it to the algorithm from that paper, as well as a baseline tailored to half-uniform distributions. Our faster algorithm achieves over **6×** reduction in the number of operations needed to correctly answer 100 random queries.

  In Figure 1, we illustrate the trade-off between the number of samples and the query time exponent $\rho_q$ in our upper and lower bounds.

- **Open Questions:** The direct question left open by our work is whether there exists a data structure whose upper bounds for general distributions match our lower bounds (note we give matching

---

[3]See Section 2 for the formal definition. Generally, proving data structure lower bounds requires restriction to a specific model of computation, and the list-of-points model is a standard choice for related approximate nearest neighbor problems.

upper bounds for half-uniform distributions). [1] give an algorithm for the general case, but with a worse trade-off than that described by our lower bound. More generally, are there other data structures problems for which one can show statistical-computational tradeoffs between the trifecta of samples, query time, and space?

## 2  Preliminaries and Roadmap for the Lower Bound

First we introduce helpful notation used throughout the paper.

**Notation:**  We use $\mathrm{Bern}(p)$ to denote the Bernoulli($p$) distribution and $\mathrm{Poi}(\lambda)$ to denote the Poisson($\lambda$) distribution. For a discrete distribution $f : X \to \mathbb{R}$, we use $supp(f) = \{x \in X : f(x) \neq 0\}$ to denote $f$'s support and $|supp(f)|$ to denote the size of its support. We use $f^n$ to denote the tensor product of $n$ identical distribution $f$. We call a distribution $f$ half-uniform if it is a uniform distribution on its support $T$ with $|T| = n/2$. For a binary distribution $P$ supported on $\{0, 1\}$ with a random variable $x \sim P$, we sometimes explicitly write $P = \begin{bmatrix} \mathbb{P}[x=1] \\ \mathbb{P}[x=0] \end{bmatrix}$. Similarly, for a joint distribution $PQ$ over $\{0,1\}^2$ with $(x, y) \sim PQ$, we write $PQ = \begin{bmatrix} \mathbb{P}[x=1, y=1] & \mathbb{P}[x=1, y=0] \\ \mathbb{P}[x=0, y=1] & \mathbb{P}[x=0, y=0] \end{bmatrix}$.

For a vector $x \in \mathbb{R}^n$, we use $x[i]$ to denote its $i$-th coordinate. We use $d(p||q) = p \log \frac{p}{q} + (1 - p) \log \frac{1-p}{1-q}$ and $D(P||Q) = \sum_{p \in P} p \log \frac{p}{q}$ to denote the standard KL-divergence over a binary distribution or a general discrete distribution. KL divergence $D(P||Q)$ is only finite when $supp(P) \subseteq supp(Q)$, also denoted as $P \ll Q$. All logarithms are natural.

We now introduce the main problem which we use to prove our statistical-computational lower bound. We state a version which generalizes half-uniform distributions.

**Definition 2.1** (Uniform random density estimation problem). For a universe $U = \{0, 1\}^n$, we generate the following problem instance:

1. A dataset $P$ is constructed by sampling $k$ uniform distributions, where for each uniform distribution $p \in P$, every element $i \in [n]$ is contained in $p$'s support with probability $w_u$.

2. Fix a distribution $p^* \in P$, take $\mathrm{Poi}\left(\frac{|supp(p^*)|}{s \cdot w_u}\right)$ samples from $p^*$ and get a query set $q$.

3. The goal of the data structure is to preprocess $P$ such that when given the query set $q$, it recovers the distribution $p^*$.

We denote this problem as $\mathrm{URDE}(w_u, s)$. URDE abbreviates *Uniform Random Density Estimation*. The name comes from the fact that the data set distributions are uniform over randomly generated supports. In Section 3, we prove a lower bound for URDE by showing that a previously studied '*hard*' problem can be reduced to URDE. The previously studied hard problem is the GapSS problem.

**Definition 2.2** (Random GapSS problem [3]). For a universe $U = \{0, 1\}^n$ and parameters $0 < w_q < w_u < 1$, let distribution $P_U = \mathrm{Bern}(w_u)^n$, $P_Q = \mathrm{Bern}(w_q)^n$, and $P_{QU} = \begin{bmatrix} w_q & 0 \\ w_u - w_q & 1 - w_u \end{bmatrix}^n$. A random $\mathrm{GapSS}(w_u, w_q)$ problem is generated by the following steps:

1. A dataset $P \subseteq U$ is constructed by sampling $k$ points where $p \sim P_U$ for all $p \in P$.

2. A dataset point $p^* \in P$ is fixed and a query point $q$ is sampled such that $(q, p^*) \sim P_{QU}$.

3. The goal of the data structure is to preprocess $P$ such that it recovers $p^*$ when given the query point $q$.

We denote this problem as random $\mathrm{GapSS}(w_u, w_q)$. GapSS abbreviates *Gap Subset Search*. To provide some intuition about how GapSS relates to URDE, let us denote the data set $P = \{p_1, \ldots, p_k\}$. Then the $p_i \in \{0, 1\}^n$ can naturally be viewed as $k$ independently generated random subsets of $[n]$. For each $i$, $p_i$ includes each element of $[n]$ with probability $w_u$. The query point $q$ can similarly be viewed as a random subset of $[n]$ including each element with probability $w_q$, but it is correlated with some fixed $p^* \in P$. Namely, $p^*$ and $q$ are generated according to the join distribution $P_{QU}$ (with the

right marginal distributions $P_Q$ and $P_U$) such $q$ a subset of $p^*$. The goal in GapSS is to identify $p^*$ given $q$. This intuition is formalized in Section 3.

Our main goal is to study the asymptotic behavior of algorithms with sublinear samples, or specifically, the query time and memory trade-off when only sublinear samples are available, so all our theorems assume the setting that both the support size $n$ and the number of samples $k$ goes to infinity and $n \ll k \leq \text{poly}(n)$. Sublinear samples mean that $\frac{1}{s} < o(1)$ as $n$ goes to infinity.

Our lower bound extend and generalize lower bounds for GapSS in the 'List-of-points' model. Thus, the lower bound we provide for URDE is also in the "List-of-points" model defined below (slightly adjusted from the original definition in [5] to our setting). The model captures a large class of data structures for retrieval problems such as partial match and nearest neighbor search: where one preprocesses a dataset $P$ to answer queries $q$ that can "match" a point in the dataset.

**Definition 2.3** (List-of-points model). Fix a universe $Q$ of queries, a universe $U$ of dataset points, as well as a partial predicate $\phi : Q \times S \to \{0, 1, \bot\}$. We first define the following $\phi$-retrieval problem: preprocess a dataset $P \subseteq U$ so that given a query $q \in Q$ such that there exist some $p^* \in P$ with $\phi(q, p^*) = 1$ and $\phi(q, p) = 0$ on all $p \in P \setminus \{p^*\}$, we must report $p^*$.

Then a list-of-points data structure solving the above problem is as follows:

1. We fix (possibly random) sets $A_i \subseteq U$, for $1 \leq i \leq m$; and with each possible query point $q \in Q$, we associate a (random) set of indices $I(q) \subseteq [m]$;

2. For the given dataset $P \subset U$, we maintain $m$ lists of points $L_1, L_2, ..., L_m$, where $L_i = P \cap A_i$.

3. On query $q \in Q$, we scan through lists $L_i$ where $i \in I(q)$, and check whether there exists some $p \in L_i$ with $\phi(q, p) = 1$. If it exists, return $p$.

The data structure succeeds, for a given $q \in Q$, $p^* \in P$ with $\phi(q, p^*) = 1$, if there exists $i \in I(q)$ such that $p^* \in L_i$. The total space is defined by $S = m + \sum_{i \in [m]} |L_i|$ and the query time by $T = |I(q)| + \sum_{i \in I(q)} |L_i|$.

To see how the lower bound model relates to URDE, in our setting, the '$\phi$-retrieval problem' is the URDE problem: $U$ is the set of random half-uniform distributions, $Q$ is the family of query samples, and $\phi(q, p)$ is 1 if the samples $q$ were drawn from the distribution $p$, and 0 otherwise. (The $\bot$ case corresponds to an "approximate" answer, considering by the earlier papers; but we define URDE problem directly to not have approximate solutions.)

We use the list-of-points model as it captures all known "data-independent" similarity search data structures, such as Locality-Sensitive Hashing [14]. In principle, a lower bound against this model does not rule out *data-dependent* hashing approaches. However, these have been useful only for datasets which are not chosen at random. In particular, [5] conjecture that data-dependency doesn't help on random instances, which is the setting of our theorems.

## 3 Lower bounds for random half-uniform density estimation problem

In this section, we formalize our lower bound. The main theorem of the section is the following.

**Theorem 3.1** (Lower bound for URDE). *If a list-of-points data structure solves the* URDE $\left(\frac{1}{2}, s\right)$ *using time* $O(k^{\rho_q})$ *and space* $O(k^{1+\rho_u})$, *and succeeds with probability at least* 0.99, *then for sufficiently large $s$, $\rho_q \geq 1 - \frac{1}{s^{1-\log 2 - o(1)}} - \frac{\rho_u}{\log s - 1}$.*

To prove Theorem 3.1, our starting point is the following result of [3] that provides a lower-bound for the random GapSS problem.

**Theorem 3.2** (Lower bound for random GapSS, [3]). *Consider any list-of-points data structure for solving the random* $\text{GapSS}(w_u, w_q)$ *problem on $k$ points, which uses expected space* $O(k^{1+\rho_u})$, *has expected query time* $O(k^{\rho_q - o_k(1)})$, *and succeeds with probability at least* 0.99. *Then for every $\alpha \in [0, 1]$, we have that*

$$\alpha \rho_q + (1 - \alpha) \rho_u \geq \inf_{\substack{t_q, t_u \in [0,1] \\ t_u \neq w_u}} F(t_u, t_q),$$

*where* $F(t_u, t_q) = \alpha \frac{D(T||P) - d(t_q||w_q)}{d(t_u||w_u)} + (1-\alpha) \frac{D(T||P) - d(t_u||w_u)}{d(t_u||w_u)}$, $P = \begin{bmatrix} w_q & 0 \\ w_u - w_q & 1 - w_u \end{bmatrix}$ *and*
$T = \underset{\substack{T \ll P \\ \mathbb{E}_{X \sim T}[X] = \begin{bmatrix} t_q \\ t_u \end{bmatrix}}}{\arg\inf} D(T||P).$

Our proof strategy is to first give a reduction from the the GapSS problem to the URDE problem. Note that the URDE problem involves a *statistical* step where we receive samples from an unknown distribution (our query). On the other hand, the query of GapSS is a specified vector, rather than a distribution, with no ambiguity. Our reduction bridges this and shows that GapSS is a 'strictly easier' problem than URDE.

**Theorem 3.3** (Reduction from random GapSS to URDE)**.** *If a list-of-points data structure solves the* URDE$(w_u, s)$ *problem of size $k$ in Definition 2.1 using time $O(k^{\rho_q})$ and space $O(k^{1+\rho_u})$, and succeeds with probability at least $0.99$, then there exists a list-of-points data structure solving the* GapSS$(w_u, w_q)$ *problem for $w_q = w_u \left(1 - e^{\frac{-1}{s \cdot w_u}}\right)$ using space $O(k^{1+\rho_u})$ and time $O(k^{\rho_q} + w_q \cdot n)$, and succeeds with probability at least $0.99$.*

*Proof.* We provide a reduction from random GapSS$(w_u, w_q)$ to URDE$(w_u, s)$ with $s = \frac{-1}{w_u \log\left(1 - \frac{w_q}{w_u}\right)}$. Specifically, for each instance $(P_1, p_1^*, q_1)$ generated from GapSS$(w_u, w_q)$ in Definition 2.2, we will construct an instance $(P_2, p_2^*, q_2)$ generated from URDE$(w_u, s)$ satisfying Definition 2.1 for some $s$.

For each point $p_1 \in P_1$, it is straightforward to construct a corresponding uniform distribution $p_2$ supported on those coordinates where $p_1[i] = 1$. Then let's construct $q_2$ from $q_1$. Recall that for each $i \in U$ with $p_1^*[i] = 1$, we have $q_1[i] = 0$ with probability $1 - \frac{w_q}{w_u}$, in which case we add no element $i$ to $q_2$. If $q_1[i] = 1$, we add $\text{Poi}_+ \left(\frac{1}{s \cdot w_u}\right)$ copies of element $i$ to $q_2$ where $\mathbb{P}\left[\text{Poi}_+(\lambda) = x\right] = \frac{\mathbb{P}[\text{Poi}(\lambda) = x]}{\mathbb{P}[\text{Poi}(\lambda) > 0]}$ for any $x > 0$. By setting $s = \frac{-1}{w_u \log\left(1 - \frac{w_q}{w_u}\right)}$, we have $\mathbb{P}\left[\text{Poi}\left(\frac{1}{s \cdot w_u}\right) = 0\right] = 1 - \frac{w_q}{w_u}$. Thus for each element $i$ in $p_2^*$, the number of its appearances in $q_2$ exactly follows the distribution $\text{Poi}(\frac{1}{s \cdot w_u})$. According to the property of the Poisson distribution, uniformly sampling $\text{Poi}\left(\frac{|supp(p_2^*)|}{s \cdot w_u}\right)$ elements from a set of size $|supp(p_2^*)|$ is equivalent to sampling each element $\text{Poi}(\frac{1}{s \cdot w_u})$ times. Therefore, the constructed instance $(P_2, p_2^*, q_2)$ is an instance of URDE$(w_u, s)$, as stated in Definition 2.1. Equivalently, we have the relationship $w_q = w_u \left(1 - e^{\frac{-1}{s \cdot w_u}}\right)$.

Hence we complete our reduction from GapSS(Definition 2.2) to URDE (Definition 2.1). $\square$

To get the desired space-time trade-off in the sublinear sample regime, which means $s \to \infty$ (or equivalently $w_q \to 0$), and to get an interpretable analytic bound, we need to develop upon the lower bound in Theorem 3.2. This requires explicitly solving the optimization problem in Theorem 3.2. Proving Theorem 3.4 (proof in Appendix 3) is the main technical contribution of the paper.

**Theorem 3.4** (Explicit lower bound for random GapSS instance)**.** *Consider any list-of-points data structure for solving the random* GapSS $\left(\frac{1}{2}, w_q\right)$ *which has expected space $O(k^{1+\rho_u})$, uses expected query time $O\left(k^{\rho_q - o(1)}\right)$, and succeeds with probability at least $0.99$. Then we have the following lower bound for sufficiently small $w_q$: $\rho_q \geq 1 - w_q^{1 - \log 2 - o(1)} + \frac{\rho_u}{1 + \log w_q}$.*

Applying our reduction to the random GapSS lower bound above allows us to prove our main theorem.

*Proof of Theorem 3.1.* According to the reduction given in Theorem 3.3 from GapSS$(w_u, w_q)$ to URDE$(w_u, s)$ where $w_q = w_u \left(1 - e^{\frac{-1}{w_u s}}\right) \geq \frac{1}{s}$. We can apply the lower bound in Theorem 3.4 and get the desired lower bound. $\square$

*Remark* 3.5. Note that in URDE $\left(\frac{1}{2}, s\right)$, the distributions are uniform over random subsets of expected size $n/2$ and the query set is generated by taking $\text{Poi}\left(\frac{2|supp(p^*)|}{s}\right)$ samples from one of them $p^*$. This is not quite saying that the query complexity is $n/s$. However, by standard

concentration bounds, from the Poisson sample, we can simulate sampling with a fixed number of samples $n/s - \tilde{O}(\sqrt{n/s}) = n/s(1 - o(1))$ with high probability, and so, any algorithm using this fixed number of samples must have the same lower bound on $\rho_q$ as in Theorem 3.1.

# 4 A simple algorithm for half-uniform density estimation problem

In this section, we present a simple algorithm for a special case of the density estimation problem when the input distributions are *half-uniform*. The algorithm also works for the related URDE($\frac{1}{2}, s$) problem of Theorem 3.1. A distribution $p$ over $[n]$ is *half-uniform* if there exists $T \subset [n]$ with $|T| = n/2$ such that $p[i] = 2/n$ if $i \in T$ and 0 otherwise. The problem we consider in this section is:

**Definition 4.1** (Half-uniform density estimation problem; HUDE($s, \varepsilon$)). For a domain $[n]$, integer $k$, $\varepsilon > 0$, and $s > 0$, we consider the following data structure problem.

1. A dataset $P$ of $k$ distributions $p_1, \ldots, p_k$ over $[n]$ which are half-uniform over subsets $T_1, \ldots, T_k \subset [n]$ each of size $n/2$ is given.

2. We receive a query set $q$ consisting of $n/s$ samples from an unknown distribution $p_{i^*} \in P$ satisfying that $\|p_{i^*} - p_j\| \geq \varepsilon$ for $j \neq i^*$.

3. The goal of the data structure is to preprocess $P$ such that when given the query set $q$, it recovers the distribution $p_{i^*}$ with probability at least 0.99.

This problem is related to the URDE($1/2, s$) problem in Theorem 3.1. Indeed, with high probability, an instance of URDE($1/2, s$) consists of *almost* half-uniform distributions with support size $n/2 \pm O(\sqrt{n \log k})$. Moreover, two such distributions $p_i, p_j$ have $\|p_i - p_j\|_1 = (1 \pm O(\sqrt{(\log k)/n}))$ with high probability. Thus, an instance of URDE($1/2, s$) is essentially an instance of HUDE($s, 1$).

To solve HUDE($s, \varepsilon$), we can essentially apply the similarity search data structure of [3] querying it with the set $Q$ consisting of all elements that were sampled at least once. This approach obtains the optimal trade-off between $\rho_u$ and $\rho_q$ (at least in the List-of-points model). The contribution of this section is to provide a *very* simple alternative algorithm with a slightly weaker trade-off between $\rho_u$ and $\rho_q$. Section 5 evaluates the simplified algorithm experimentally. Our main theorem is:

**Theorem 4.2.** *Suppose $n$ and $k$ are polynomially related, $s \geq 2$, and that $s$ is such that[4] $\frac{n}{s} \geq C \frac{\log k}{\varepsilon^2}$ for a sufficiently large constant $C$. Let $\varepsilon > 0$ and $\rho_u > 0$ be given. There exists a data structure for the HUDE($s, \varepsilon$) problem using space $O(k^{1+\rho_u} + nk)$ and with query time $O\left(k^{1 - \frac{\varepsilon \rho_u}{2 \log(2s)}} + n/s\right)$.*

Let us compare the upper bound of Theorem 4.2 to the lower bound in Theorem 3.1. While Theorem 4.2 is stated for half-uniform distributions, its proof is easily modified to work for the URDE($1/2, s$) problem where the support size is random. Then $\varepsilon = 1 - O(1)$ and as $s \to \infty$, $\frac{2}{1 - e^{-2/s}} = s(1 + o(1))$. Thus, the query time exponent in Theorem 4.2 is $\rho_q = 1 - (1 + o(1)) \frac{\log(2) \rho_u}{\log s}$. URDE($1/2, s$) is exactly the hard instance in Theorem 3.1, and so we know that any algorithm must have $\rho_q \geq 1 - (1 + o(1)) \frac{\rho_u}{\log s}$ as $s \to \infty$. Asymptotically, our algorithm therefore gets the right logarithmic dependence on $s$ but with a leading constant of $\log(2) \approx 0.693$ instead of 1.

Next we define the algorithm. Let $\ell$ and $L$ be parameters which we will also specify shortly. During preprocessing, our algorithm samples $L$ subsets $S_1, \ldots, S_L$ of $[n]$ each of size $\ell$ independently and uniformly at random. For each $i \in L$, it stores a set $A_i$ of all indices $j \in [k]$ such that $S_i \subset T_j$, namely the indices of the distributions $p_j$ which contain $S_i$ in their support. See Algorithm 1.

During the query phase, we receive $n/s$ samples from $p = p_{i^*}$ for some unknown $i^* \in [k]$. Our algorithm first forms the subset $Q \subset [n]$ consisting of all elements that were sampled at least once. Note that $|Q| \leq n/s$ as elements can be sampled several times. The algorithm proceeds in two steps. First, it goes through the $L$ sets $S_1, \ldots, S_L$ until it finds an $i$ such that $S_i \subset Q$. Second, it scans through the indices $j \in A_i$. For each such $j$ it samples a set $U_j$ one element at a time from $Q$. It stops this sampling at the first point of time where either $U_j \nsubseteq T_j$ or $|U_j| = C \frac{\log n}{\varepsilon}$ for a sufficiently

---

[4]The requirement $n/s \gg \log k/\varepsilon^2$ is the information theoretic lower bound for the density estimation problem.

---

**Algorithm 1** Density estimation for half-uniform distributions (preprocessing)

---

1: **Input**: Half uniform distributions $\{p_i\}_{i=1}^k$ over $[n]$ with support sets $\{T_i\}_{i=1}^k$.
2: **Output**: A data structure for the density estimation problem.
3: **procedure** PREPROCESSING($\{p_i\}_{i=1}^k$)
4:     **for** $i = 1$ to $L$ **do**
5:         $S_i \leftarrow$ sample of size $\ell$ from $[n]$
6:         $A_i \leftarrow \{j \in [k] \mid S_i \subset T_j\}$
7:     **Return** $(S_i, A_i)_{i \in [L]}$

---

---

**Algorithm 2** Query algorithm for half-uniform distributions

---

1: **Input**: Half uniform distributions $\{p_i\}_{i=1}^k$ over $[n]$ with support sets $\{T_i\}_{i=1}^k$, data structure
   $(S_i, A_i)_{i \in [L]} \leftarrow$ `Preprocessing`($\{p_i\}_{i=1}^k$), and $n/s$ samples from query distribution $p = p_{i^*}$.
2: **Output**: A distribution $p_j$.
3: **procedure** DENSITY-ESTIMATION($\{p_i\}_{i=1}^k$)
4:     $Q \leftarrow \{i \in [n] \mid i$ appeared in the $n/s$ samples$\}$
5:     **for** $i = 1$ to $L$ **do**
6:         **if** $S_i \subset Q$ **then**
7:             **for** $j \in A_i$ **do**
8:                 $U_j \leftarrow$ sample from $Q$ of size $C\frac{\log n}{\varepsilon}$ for a large constant $C$.
9:                 **if** $U_j \subset T_j$ **then**
10:                     **Return:** $p_j$

---

large constant $C$. In the first case, it concludes that $p_j$ is not the right distribution and proceeds to the next element of $A_j$ and in the latter case, it returns the distribution $p_j$ as the answer to the density estimation problem. See Algorithm 2. We defer the formal proofs to Appendix B.

## 5 Experiments

We test our algorithm in Section 4 experimentally on datasets of half-uniform distributions as in [1] and corresponding to our study in Sections 3 and 4. Given parameters $k$ and $n$, the input distributions are $k$ distributions each uniform over a randomly chosen $n/2$ elements.

**Algorithms**    We compare two main algorithms: an implementation of the algorithm presented in Section 4 which we refer to as the Subset algorithm and a baseline for half-uniform distributions which we refer to as the Elimination algorithm. The Subset algorithm utilizes the same techniques as that presented in Section 4 but with some slight changes geared towards practical usage. We do not compare to the "FastTournament" of [1] since it was computationally prohibitive; see Remark C.1.

The Subset algorithm picks $L$ subsets of size $\ell$ and preprocesses the data by constructing a dictionary mapping subsets to the distributions whose support contains that subset. When a query arrives, scan through the $L$ subset until we find one that is contained in the support of the query. We then restrict ourselves to solving the problem over the set of distributions mapped to by that subset and run Elimination. The Elimination algorithm goes through the samples one at a time. It starts with a set of distributions which is the entire input in general or a subset of the input when called as a subroutine of the Subset algorithm. To process a sample, the Elimination algorithm removes from consideration all distributions which do not contain that element in its support. When a single distribution remains, the algorithm returns that distribution as the solution. As the input distributions are random half-uniform distributions, we expect to throw away half of the distributions at each step (other than the true distribution) and terminate in logarithmically in size of the initial set of distribution steps.

**Experimental Setup**    Our experiments compare the Subset and Elimination algorithms while varying several problem parameters: the number of distributions $k$, the domain size $n$, the number of samples $S$ (for simplicity, we use this notation rather than $n/s$ samples as in the rest of the paper), and the size of subsets $\ell$ used by the Subset algorithm. While we vary one parameter at a time, the others are set to a default of $k = 50000$, $n = 500$, $S = 50$, $l = 3$. Given these parameters, we

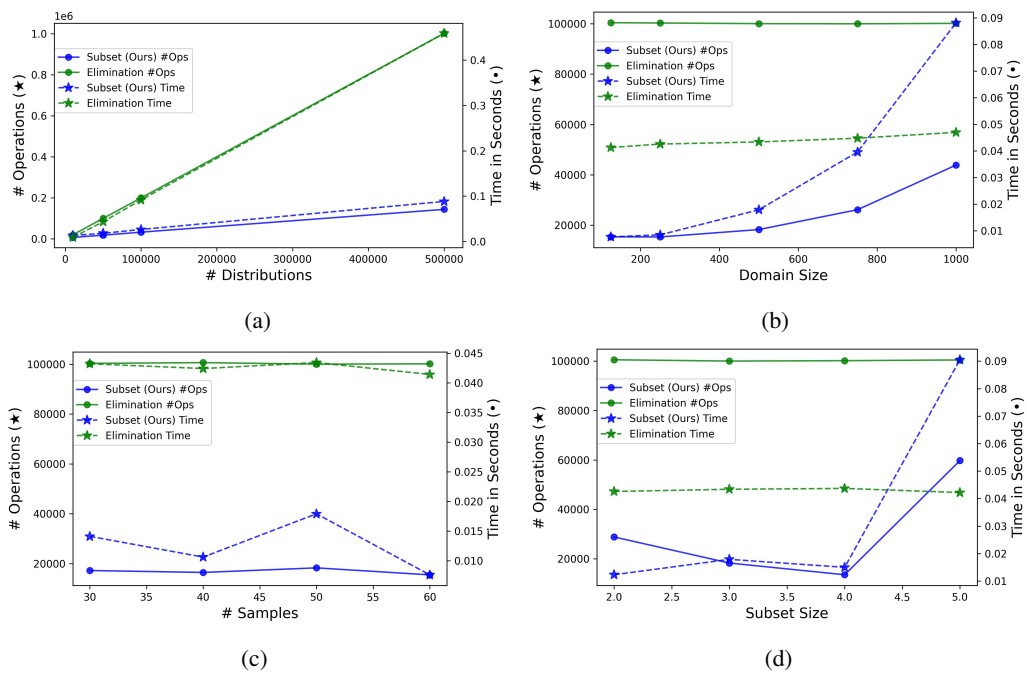

(a)             (b)

(c)             (d)

Figure 2: Comparison of efficiency of the Subset (Ours) and Elimination algorithms as (a): the number of distributions $k$ varies. Other parameters are set to $n = 500, S = 50, \ell = 3$. (b): the domain size $n$ varies. Other parameters are set to $k = 50000, S = 50, \ell = 3$. (c): the number of samples $S$ varies. Other parameters are set to $k = 50000, n = 500, \ell = 3$. (d): the subset size $\ell$ varies. Other parameters are set to $k = 50000, n = 500, S = 50$.

evaluate the Subset algorithm and the Elimination algorithm on 100 random queries where each query corresponds to picking one of the input distributions as the true distribution to draw samples from.

In all settings we test, the Elimination algorithm achieves $100\%$ accuracy on these queries, which is to be expected as long as the number of samples is sufficently more than the $\log_2 k$. There is a remaining free parameter, which is the number of subsets $L$ used in the Subset algorithm. We start with a modest value of $L = 200$ and increase $L$ by a factor of $1.5$ repeatedly until the Subset algorithm also achieves $100\%$ accuracy on the queries (in reality, it's failure probability will likely still be greater than that of the Elimination algorithm). The results we report correspond to this smallest value of $L$ for which the algorithm got all the queries correct.

For both algorithms, we track the average number of operations as well as the execution time of the algorithms (not counting preprocessing). A single operation corresponds to a membership query of checking whether a given distribution/sample contains a specified element in its support which is the main primitive used by both algorithms. We use code from [1] as a basis for our setup.

**Results** For all parameter settings we test, the number of operations per query by our Subset algorithm is significantly less than those required by Elimination algorithm, up to a factor of more than **6x**. The average query time (measured in seconds) shows similar improvements for the Subset algorithm though for some parameter settings, it takes more time than the Elimination algorithm. While, in general, operations and time are highly correlated, these instances where they differ may depend on the specific Python data structures used to implement the algorithms, cache efficiency, or other computational factors.

As the number of distributions $k$ increases, Figure 2a shows that both time and number of operations scale linearly. Across the board, our Subset algorithm outperforms the Elimination algorithm baseline and exhibits a greater improvement as $k$ increases. On the other hand, as the domain size increases in Figure 2b, the efficiency of the Subset algorithm degrades while the Elimination algorithm maintains its performance. This is due to the fact that for larger domains, more subsets are needed in order to correctly answer all queries, leading to a greater runtime.

In Figure 2c, we see that across the board, as we vary the number of samples, the Subset algorithm has a significant advantage over the Elimination algorithm in query operations and time. Finally, Figure 2d shows that for subset size $\ell \in \{2, 3, 4\}$, the Subset algorithm experiences a significant improvement over the Elimination algorithm. But for $\ell = 5$, the improvement (at least in terms of time) suddenly disappears. For this setting, that subset size requires many subsets in order to get high accuracy, leading to longer running times.

Overall, on flat distributions for a variety of parameters, our algorithm has significant benefits even over a baseline tailored for this case. The good performance of the Subset algorithm corresponds with our theory and validates the contribution of providing a simple algorithm for density estimation in this hard setting.

**Acknowledgments:** This work was supported by the Jacobs Presidential Fellowship, the Mathworks Fellowship, the NSF TRIPODS program (award DMS-2022448) and Simons Investigator Award; also supported in part by NSF (CCF2008733) and ONR (N00014-22-1-2713). Justin Chen is supported by an NSF Graduate Research Fellowship under Grant No. 174530. Shyam Narayanan is supported by an NSF Graduate Fellowship and a Google Fellowship. Anders Aamand was supported by the DFF-International Postdoc Grant 0164-00022B and by the VILLUM Foundation grants 54451 and 16582.

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

# A  Appendix: Omitted Proofs of Section 3

**Theorem 3.4** (Explicit lower bound for random GapSS instance). *Consider any list-of-points data structure for solving the random* $\mathrm{GapSS}\left(\frac{1}{2}, w_q\right)$ *which has expected space* $O(k^{1+\rho_u})$, *uses expected query time* $O\left(k^{\rho_q - o(1)}\right)$, *and succeeds with probability at least* $0.99$. *Then we have the following lower bound for sufficiently small* $w_q$: $\rho_q \geq 1 - w_q^{1 - \log 2 - o(1)} + \frac{\rho_u}{1 + \log w_q}$.

*Proof.* Our proof proceeds by explicitly calculating the lower bound given in Theorem 3.2 when $w_u = \frac{1}{2}$ and $w_q$ approaches 0. Recall that Theorem 3.2 states that if a list-of-points data structure solves $\mathrm{GapSS}(w_u, w_q)$ for $k$ points uses expected space $k^{1+\rho_u}$, and has expected query time $k^{\rho_q - o_k(1)}$, then for every $\alpha \in [0, 1]$, we have that

$$\alpha \rho_q + (1 - \alpha)\rho_u \geq \inf_{\substack{t_q, t_u \in [0,1] \\ t_u \neq w_u}} \left( \alpha \frac{D(T||P) - d(t_q||w_q)}{d(t_u||w_u)} + (1 - \alpha) \frac{D(T||P) - d(t_u||w_u)}{d(t_u||w_u)} \right) \quad (1)$$

where $P = \begin{bmatrix} w_q & 0 \\ w_u - w_q & 1 - w_u \end{bmatrix}$ and $T = \underset{\substack{T \ll P \\ \mathbb{E}_{X \sim T}[X] = \begin{bmatrix} t_q \\ t_u \end{bmatrix}}}{\arg\inf} D(T||P)$.

We denote the fraction in the right hand side of Equation 1 as $F(t_q, t_u)$. Our goal is to provide a lower bound in the case $w_u = 1/2$.

First, notice that to satisfy $T \ll P$ (i.e. $supp(T) \subseteq supp(P)$) and $\mathbb{E}_{X \sim T}[X] = \begin{bmatrix} t_q \\ t_u \end{bmatrix}$, the only available choice is $T = \begin{bmatrix} t_q & 0 \\ t_u - t_q & 1 - t_u \end{bmatrix}$. Plugging this in and expanding $F(t_u, t_q)$, we get

$$F(t_u, t_q) = \frac{(t_u - t_q) \log \frac{t_u - t_q}{w_u - w_q} + \alpha \cdot d(t_u||w_u) - t_u \log \frac{t_u}{w_u} - \alpha \cdot d(t_q||w_q) + t_q \log \frac{t_q}{w_q}}{d(t_u||w_u)}. \quad (2)$$

For $w_u = 1/2$ fixed, and for fixed $w_q, \alpha, t_u \neq 1/2$, we can consider $F$ as a function of only $t_q$. Because $F$ is a continuously differentiable function in terms of $t_q$, the infimum of $F$ (for fixed $w_u, w_q, \alpha, t_u$) can only be achieved either when $\partial F / \partial t_q = 0$, or at the boundary points ($t_q = 0$ or $t_q \to t_u^-$).

We first consider the case when the partial derivative is 0 and handle the endpoint cases later. Calculating the partial derivative of $F$ with respect to $t_q$ gives us $\frac{\partial F}{\partial t_q} = \log \frac{t_q}{w_q} - \log \frac{t_u - t_q}{w_u - w_q} - \alpha \log \frac{t_q(1 - w_q)}{w_q(1 - t_q)}$. When $\frac{\partial F}{\partial t_q} = 0$, we must have

$$\frac{t_u - t_q}{w_u - w_q} = \left( \frac{t_q}{w_q} \right)^{1 - \alpha} \left( \frac{1 - t_q}{1 - w_q} \right)^{\alpha}. \quad (3)$$

Plugging in the relation in (3) and $w_u = \frac{1}{2}$, we have

$$F(t_u, t_q) = \alpha + \frac{(t_u - t_q) \log \frac{t_u - t_q}{w_u - w_q} - t_u \log \frac{t_u}{w_u} - \alpha \cdot d(t_q||w_q) + t_q \log \frac{t_q}{w_q}}{d(t_u||w_u)}$$

$$= \alpha + \frac{t_u \left( \log \frac{t_u - t_q}{w_u - w_q} - \log \frac{t_u}{w_u} \right) - t_q \log \frac{t_u - t_q}{w_u - w_q} - \alpha \cdot d(t_q||w_q) + t_q \log \frac{t_q}{w_q}}{d(t_u||w_u)}$$

$$= \alpha + \frac{t_u \left( \log \frac{t_u - t_q}{w_u - w_q} - \log \frac{t_u}{w_u} \right) - t_q \log \frac{t_u - t_q}{w_u - w_q} + (1 - \alpha) \cdot t_q \log \frac{t_q}{w_q} - \alpha \cdot (1 - t_q) \log \frac{1 - t_q}{1 - w_q}}{d(t_u||w_u)}$$

$$= \alpha + \frac{t_u \left( \log \frac{t_u - t_q}{w_u - w_q} - \log \frac{t_u}{w_u} \right) - \alpha \log \frac{1 - t_q}{1 - w_q}}{d(t_u||w_u)} \qquad \text{(Plugging in equation 3)}$$

$$= \alpha + \frac{t_u \left( \log \left( 1 - \frac{t_q}{t_u} \right) + \log \frac{1}{1 - 2w_q} \right) - \alpha \log \frac{1 - t_q}{1 - w_q}}{d(t_u||\frac{1}{2})}. \qquad \text{(Plugging in } w_u = \frac{1}{2})$$

By Lemma A.1, if we set $\alpha = 1 + \frac{1}{\log w_q}$ for $w_q$ is sufficiently small, we have $F(t_u, t_q) \geq \alpha - w_q^{1-\log 2 - o(1)}$, uniformly over $t_u, t_q$. Next, let's check the boundary cases. For $t_q = 0$, Lemma A.8 proves that $F(t_u, 0) \geq \alpha - w_q^{1-o(1)} \geq \alpha - w_q^{1-\log 2 - o(1)}$. For $t_q \to t_u^-$, because $\frac{\partial F}{\partial t_q}$ is continuous for $0 \leq t_q < t_u$ and $\lim_{t_q \to t_u^-} \frac{\partial F}{\partial t_q} = +\infty$, the infimum of $F$ cannot be achieved when $t_q \to t_u^-$.

Thus, for $w_u = 1/2$, any fixed $w_q$ sufficiently small, $\alpha = 1 + \frac{1}{\log w_q}$, and any fixed $t_u \neq 1/2$ and the infimum of $F$ across $0 \leq t_q < t_u$ is at least $\alpha - w_q^{1-\log 2 - o(1)}$, where the $o(1)$ term goes to 0 as $w_q$ goes to 0, uniformly across $t_u, t_q$. So in fact, $F(t_u, t_q) \geq \alpha - w_q^{1-\log 2 - o(1)}$ uniformly across $t_u, t_q$. Applying this bound back to our original inequality in 1 gives us the desired bound. $\square$

Our goal is to now bound the fraction in the final step of the proof of Theorem 3.4. The following lemma bound this fraction.

**Lemma A.1.** *Fix any constant $\delta > 0$. Suppose $w_q < 1$ is smaller than a sufficiently small constant $c = c_\delta$ that only depends on $\delta$, $w_u = 1/2$, and $\alpha = 1 + \frac{1}{\log w_q}$. Suppose these parameters satisfy the relation given in Equation 3. Then*

$$\inf_{\substack{t_q, t_u \in [0,1] \\ t_u \neq w_u}} \frac{t_u \left( \log \left( 1 - \frac{t_q}{t_u} \right) + \log \frac{1}{1-2w_q} \right) - \alpha \log \frac{1-t_q}{1-w_q}}{d(t_u \| w_u)} \geq -w_q^{1-\log 2 - \delta}.$$

*Equivalently,*

$$\inf_{\substack{t_q, t_u \in [0,1] \\ t_u \neq w_u}} \frac{t_u \left( \log \left( 1 - \frac{t_q}{t_u} \right) + \log \frac{1}{1-2w_q} \right) - \alpha \log \frac{1-t_q}{1-w_q}}{d(t_u \| w_u)} \geq -w_q^{1-\log 2 - o(1)}$$

*for sufficiently small $w_q$, where $o(1)$ denotes a term that uniformly goes to 0 as $w_q \to 0$ (regardless of $t_q, t_u$).*

In the rest of this section, we use $o(1)$ to denote any term that goes to 0 as $w_q \to 0$, uniformly over $t_q, t_u$. We will prove some auxiliary lemmas before proving Lemma A.1. We first define the following function $H(x)$.

**Definition A.2.** For a value $x$, $H(x) := x \log(2x) + (1-x) \log(2(1-x))$.

It is clear that $H(x)$ is only defined when $0 < x < 1$. Moreover, we note the following basic property and provide its proof for completeness.

**Proposition A.3.** *For any $x \in (0, 1)$, $2(\frac{1}{2} - x)^2 \leq H(x) \leq 16(\frac{1}{2} - x)^2$.*

*Proof.* It is simple to check that $H(1/2) = 0$ and $H'(1/2) = 0$. Moreover, the second derivative is $H''(x) = \frac{1}{x} + \frac{1}{1-x} = \frac{1}{x-x^2}$. For $0 < x < 1$, $x - x^2 \leq 1/4$, so $H''(x) \geq 4$ for all $x$. Thus, $H''(x) \geq 2(\frac{1}{2} - x)^2$.

Next, we have that $x - x^2 \geq \frac{3}{16}$ for $x \in [1/4, 3/4]$, which means $H''(x) \leq \frac{16}{3}$ for $x \in [1/4, 3/4]$. Thus, $H(x) \leq \frac{8}{3}(\frac{1}{2} - x)^2$ for $x \in [1/4, 3/4]$. Since the first derivative $H'(x) = \log(x) - \log(1-x)$ is negative for $x < \frac{1}{2}$ and positive for $x > \frac{1}{2}$, this means $H(x)$ is maximized as $x$ approaches either 0 or 1. But the limits equal $\log 2$, so $H(x) \leq \log 2$ for all $x$. Since $(\frac{1}{2} - x)^2 \geq \frac{1}{16}$ for all $1 > x > \frac{3}{4}$ or $0 < x < \frac{1}{4}$, this means for any such $x$, $H(x) \leq \log 2 \leq 16 \log 2 \cdot (\frac{1}{2} - x)^2 \leq 16 \cdot (\frac{1}{2} - x)^2$. So in either case, $H(x) \leq 16(\frac{1}{2} - x)^2$. $\square$

We are now ready to prove Lemma A.1.

*Proof of Lemma A.1.* For simplicity of notation, let $x = t_q$. Recalling the value of $\alpha$, $w_u$, and Equation 3, we have

$$t_u = x + \left( \frac{x}{w_q} \right)^{1-\alpha} \cdot \left( \frac{1-x}{1-w_q} \right)^{\alpha} \cdot \left( \frac{1}{2} - w_q \right).$$

Now we denote the fraction in the lemma statement as $b(x)/a(x)$ and we note

$$a = H(t_u) = t_u \cdot \log(2t_u) + (1 - t_u) \cdot \log(2(1 - t_u)),$$

$$b = -\alpha \cdot \log\left(\frac{1-x}{1-w_q}\right) + t_u \cdot \left(\log\left(1 - \frac{x}{t_u}\right) + \log\left(\frac{1}{1 - 2w_q}\right)\right). \tag{4}$$

From Proposition A.4, it suffices to check the following cases:

1. $0 < x \le w_q^{1.01}$, or $w_q^{0.99} \le x < x^*$,

2. $w_q^{1.01} < x < (1 + \frac{1}{\log w_q})w_q$, or $(1 - \frac{1}{\log w_q})w_q < x < w_q^{0.99}$,

3. and $(1 + \frac{1}{\log w_q})w_q \le x \le (1 - \frac{1}{\log w_q})w_q$.

In Case 1, $x^*$ is such that $x \in (0, x^*)$ is the regime of $x$ such that $a$ and $b$ are well defined. Proposition A.4 further states that $x^*$ only depends on $w_q$ and $x^* \le w_q^{1-\log 2 - o(1)}$ as $w_q \to 0$.

The cases are handled in Lemmas A.5, A.6, and A.7, respectively. Combining them proves Lemma A.1. □

**Proposition A.4.** *Suppose that $0 < w_q < \frac{1}{3}$ and $0 \le x \le 1$. Then, the regime of $x$ such that $a, b$ are well defined is $x \in (0, x^*)$, where $0 < x^* < 1$ only depends on $w_q$ and $x^* \le w_q^{1-\log 2 - o(1)}$ as $w_q \to 0$.*

*Proof.* We must have that $0 < t_u < 1$, so that $a = H(t_u)$ is well-defined. If $x = 0$, $t_u = 0$, and if $x = 1$, then $\log\left(\frac{1-x}{1-w_q}\right)$ isn't defined, so $b$ isn't defined. If $0 < x < 1$, $t_u$ is always positive, so $a$ being well-defined is equivalent to $1 - t_u > 0$, which is equivalent to

$$1 - x > \left(\frac{x}{w_q}\right)^{1-\alpha} \cdot \left(\frac{1-x}{1-w_q}\right)^{\alpha} \cdot \left(\frac{1}{2} - w_q\right).$$

We can rearrange this as

$$\left(\frac{1-x}{x}\right)^{-1/(\log w_q)} = \left(\frac{1-x}{x}\right)^{1-\alpha} > \frac{1/2 - w_q}{w_q^{1-\alpha}(1 - w_q)^{\alpha}} = \frac{e(1/2 - w_q)}{(1 - w_q)^{1+1/(\log w_q)}}.$$

We can again rearrange this as

$$\frac{1-x}{x} > C(w_q) := \left(\frac{e(1/2 - w_q)}{(1 - w_q)^{1+1/(\log w_q)}}\right)^{-\log w_q},$$

where $C(w_q)$ is positive if $w_q < 1/3$. This is equivalent to is equivalent to $0 < x < \frac{1}{C(w_q)+1}$. So, we can set $x^* = \frac{1}{C(w_q)+1}$. Thus, $a$ is well defined if and only if $0 < x < x^*$, where $x^* \in (0, 1)$ clearly holds. Moreover, if $w_q < 1/3$, then $\log\frac{1-x}{1-w_q}$ and $\log\frac{1}{1-2w_q}$ are well-defined, and $t_u > x > 0$ so $\log\left(1 - \frac{x}{t_u}\right)$ is also well-defined. In summary, $a, b$ are well defined if and only if $0 < x < x^* = \frac{1}{1+C(w_q)}$.

Finally, we bound $x^*$ for $w_q$ sufficiently small. Note that $(1 - w_q)^{1+1/(\log w_q)} = 1 + o(1)$ and $1/2 - w_q = 1/2 - o(1)$, so $C(w_q) = \left(\frac{e}{2} \cdot (1 \pm o(1))\right)^{-\log w_q} = (e/2)^{-\log w_q \cdot (1 \pm o(1))} = w_q^{\log 2 - 1 \pm o(1)}$. Thus, $C(w_q) \ge w_q^{\log 2 - 1 + o(1)}$, which means that $x^* \le w_q^{1-\log 2 - o(1)}$. □

**Lemma A.5.** *Suppose that $w_q$ is sufficiently small, and $0 < x \le w_q^{1.01}$, or $w_q^{0.99} \le x < x^*$. Then, $\frac{b}{a} \ge -w_q^{1-\log 2 - o(1)}$ as $w_q \to 0$.*

*Proof.* Since $x$ is strictly positive, we can write $x = w_q^{1+\gamma}$, for some real $\gamma$ with $|\gamma| \ge 0.01$. Then, $\frac{x}{w_q} = w_q^{\gamma}$, so $\left(\frac{x}{w_q}\right)^{1-\alpha} = w_q^{-\gamma/\log w_q} = e^{-\gamma}$. Finally, since $x = o(1)$ and $w_q = o(1)$, this means

$t_u = x + e^{-\gamma} \cdot \left(\frac{1}{2} \pm o(1)\right) = 0.5 \cdot e^{-\gamma} \pm o(1) \cdot (e^{-\gamma} + 1)$. Since $|\gamma| > 0.01$, this implies that, for $w_q$ sufficiently small, $|t_u - 0.5| \geq \Omega(1)$, which means $a = H(t_u) = \Omega(1)$ by Proposition A.3.

We can also bound $b$ as follows. Note that $\left|\log\left(\frac{1-x}{1-w_q}\right)\right| = O(x + w_q)$, so $-\alpha \cdot \log\left(\frac{1-x}{1-w_q}\right) \geq -O(x + w_q)$. Next, note that for $w_q$ sufficiently small (and thus $x < x^*$ is also sufficiently small), since $0 < \alpha, 1 - \alpha < 1$, $\left(\frac{1-x}{1-w_q}\right)^{\alpha} \geq 0.5$, and $\left(\frac{x}{w_q}\right)^{1-\alpha} \geq x^{1-\alpha} \geq x$. Also, $\frac{1}{2} - w_q \geq \frac{1}{3}$. Thus, $t_u \geq x + x \cdot \frac{1}{2} \cdot \frac{1}{3} \geq \frac{7}{6} \cdot x$. Therefore, $0 \leq \frac{x}{t_u} \leq \frac{6}{7}$, which means $\left|\log\left(1 - \frac{x}{t_u}\right)\right| \leq O(x/t_u)$. Thus, $\left|t_u \cdot \left(\log\left(1 - \frac{x}{t_u}\right) + \log\left(\frac{1}{1-2w_q}\right)\right)\right| \leq O(t_u \cdot (x/t_u + w_q)) = O(x + w_q)$. Thus, $|b| \leq O(x + w_q)$.

Since $a = \Omega(1)$ is positive, and $b \geq -O(x + w_q)$, this means that $\frac{b}{a} \geq -O(x + w_q)$. Since we know that $x < x^* \leq w_q^{1 - \log 2 - o(1)}$, this implies that $\frac{b}{a} \geq -w_q^{1 - \log 2 - o(1)}$. $\qquad\square$

**Lemma A.6.** *Suppose that* $w_q^{1.01} < x < (1 + \frac{1}{\log w_q})w_q$, *or* $(1 - \frac{1}{\log w_q})w_q < x < w_q^{0.99}$. *Then,* $\frac{b}{a} \geq -w_q^{0.99 - o(1)}$.

*Proof.* We again write $x = w_q^{1 + \gamma}$, where this time $|\gamma| < 0.01$. Since either $x > (1 - \frac{1}{\log w_q})w_q$ or $x < (1 + \frac{1}{\log w_q})w_q$, this means that $|\gamma| \geq \Omega\left(\frac{1}{(\log w_q)^2}\right)$. Then, we can write

$$t_u = O(w_q^{0.99}) + w_q^{-\gamma/(\log w_q)} \cdot \left(\frac{1}{2} \pm O(w_q^{0.99})\right) = \frac{e^{-\gamma}}{2} \pm O(w_q^{0.99}) = \frac{1}{2} - \Theta(\gamma),$$

since $0.01 > |\gamma| \geq \Omega\left(-\frac{1}{\log w_q}\right)$ so the $w_q^{0.99}$ term is negligible compared to $\gamma$. So, $a = H(t_u) = \Theta(\gamma^2) \geq \Omega\left(1/(\log w_q)^4\right)$, by Proposition A.3.

Next, to bound $b$, note that $\left|\log\left(\frac{1-x}{1-w_q}\right)\right| = O(x + w_q) \leq O(w_q^{0.99})$. Also, since $t_u = \frac{e^{-\gamma}}{2} \pm O(w_q^{0.99})$, this means that $t_u = \Theta(1)$, so $\left|t_u \cdot \left(\log\left(1 - \frac{x}{t_u}\right) + \log\left(\frac{1}{1-2w_q}\right)\right)\right| \leq O(x + w_q) = O(w_q^{0.99})$. Overall, $|b| \leq O(w_q^{0.99})$, so $\frac{b}{a} \geq -w_q^{0.99 - o(1)}$. $\qquad\square$

**Lemma A.7.** *Suppose that* $(1 + \frac{1}{\log w_q})w_q \leq x \leq (1 - \frac{1}{\log w_q})w_q$. *Then,* $\frac{b}{a} \geq -w_q^{1 - o(1)}$.

*Proof.* Suppose that $x = (1 + \beta)w_q$, where $|\beta| \leq -\frac{1}{\log w_q}$. We will look at $t_u$ from a more fine grained perspective.

Note that $\left(\frac{x}{w_q}\right)^{1-\alpha} = (1 + \beta)^{-1/(\log w_q)} = e^{-(\beta \pm O(\beta^2))/(\log w_q)} = 1 - \frac{\beta}{\log w_q} \pm O\left(\frac{\beta^2}{\log w_q}\right)$. Moreover, $\left(\frac{1-x}{1-w_q}\right)^{\alpha} = \left(1 - \frac{\beta w_q}{1-w_q}\right)^{\alpha} = 1 - \frac{\beta w_q}{1-w_q} \cdot \alpha \pm O(\beta^2 w_q^2)$. Thus,

$$\left(\frac{x}{w_q}\right)^{1-\alpha} \cdot \left(\frac{1-x}{1-w_q}\right)^{\alpha} = 1 - \frac{\beta}{\log w_q} - \frac{\beta w_q}{1-w_q} \cdot \alpha \pm O\left(\frac{\beta^2}{\log w_q}\right).$$

Thus, we can write

$$t_u = w_q + \beta w_q + \left(1 - \frac{\beta}{\log w_q} - \frac{\beta w_q}{1-w_q} \cdot \alpha \pm O\left(\frac{\beta^2}{\log w_q}\right)\right) \cdot \left(\frac{1}{2} - w_q\right)$$

$$= \frac{1}{2} + \beta w_q - \beta \cdot \left(\frac{1}{\log w_q} + \frac{w_q}{1-w_q} \cdot \alpha\right) \cdot \left(\frac{1}{2} - w_q\right) \pm O\left(\frac{\beta^2}{\log w_q}\right) \qquad (5)$$

Note that $t_u = \frac{1}{2} - \Theta(\beta/\log w_q)$, since the other terms in (5) are all negligible compared to $\beta/\log w_q$, which means that $a = \Theta(\beta^2/(\log w_q)^2)$ by Proposition A.3.

To bound $b$, we will need the more fine-grained approximation of $t_u$ from (5). Indeed, note that $\log\left(\frac{1-x}{1-w_q}\right) = \log\left(1 - \frac{\beta w_q}{1-w_q}\right) = -\frac{\beta w_q}{1-w_q} \pm O(\beta^2 w_q^2)$, and $t_u \cdot \left(\log\left(1 - \frac{x}{t_u}\right) + \log\left(\frac{1}{1-2w_q}\right)\right) =$

$t_u \cdot \log\left(\frac{t_u - x}{t_u(1 - 2w_q)}\right) = t_u \cdot \log\left(1 + \frac{2w_q t_u - x}{t_u(1 - 2w_q)}\right)$. Note that $2w_q t_u = 2(1/2 - \Theta(\beta/\log w_q))w_q = w_q - \Theta(\beta \cdot w_q/\log w_q)$. Thus, $2w_q t_u - x = \Theta(\beta \cdot w_q)$. Moreover, $t_u(1 - 2w_q) = \Theta(1)$. Therefore, we have that

$$t_u \cdot \log\left(1 + \frac{2w_q t_u - x}{t_u(1 - 2w_q)}\right) = t_u \cdot \left(\frac{2w_q t_u - x}{t_u(1 - 2w_q)} \pm O\left(\frac{2w_q t_u - x}{t_u(1 - 2w_q)}\right)^2\right) = \frac{2w_q t_u - x}{1 - 2w_q} \pm O(\beta^2 w_q^2).$$

Therefore,

$$b = \alpha \cdot \frac{\beta w_q}{1 - w_q} + \frac{2w_q t_u - x}{1 - 2w_q} \pm O(\beta^2 w_q^2).$$

Using the more fine-grained approximation of $t_u$, we have that

$$2w_q t_u - x = w_q + 2\beta w_q^2 - w_q \beta \left(\frac{1}{\log w_q} + \frac{w_q}{1 - w_q} \cdot \alpha\right) \cdot (1 - 2w_q) - (w_q + \beta w_q) \pm O(\beta^2 w_q)$$

$$= (2\beta w_q^2 - \beta w_q) - w_q \beta \left(\frac{1}{\log w_q} + \frac{w_q}{1 - w_q} \cdot \alpha\right) \cdot (1 - 2w_q) \pm O(\beta^2 w_q)$$

$$= -w_q \beta(1 - 2w_q) - w_q \beta \left(\frac{1}{\log w_q} + \frac{w_q}{1 - w_q} \cdot \alpha\right) \cdot (1 - 2w_q) \pm O(\beta^2 w_q)$$

$$= w_q \beta \cdot \left(-1 - \frac{1}{\log w_q} - \frac{w_q}{1 - w_q} \cdot \alpha\right) \cdot (1 - 2w_q) \pm O(\beta^2 w_q)$$

$$= -w_q \beta \cdot \left(\frac{\alpha}{1 - w_q}\right) \cdot (1 - 2w_q) \pm O(\beta^2 w_q),$$

where the last line uses the fact that $\alpha = 1 + \frac{1}{\log w_q}$. So,

$$b = \alpha \cdot \frac{\beta w_q}{1 - w_q} - w_q \beta \cdot \frac{\alpha}{1 - w_q} \pm O(\beta^2 w_q) = \pm O(\beta^2 w_q).$$

Therefore, $\left|\frac{b}{a}\right| \leq O(w_q \cdot (\log w_q)^2) \leq w_q^{1 - o(1)}$. $\qquad \square$

Finally we check the endpoint cases of Theorem 3.4.

**Lemma A.8.** *For $F(t_u, t_q)$ defined in Equation 2, and for $\alpha = 1 + \frac{1}{\log w_q}$, $F(t_u, 0) \geq \alpha - w_q^{1 - o(1)}$ where $o(1)$ goes to 0 as $w_q$ goes to 0, uniformly over $t_u$.*

*Proof.* For $t_q = 0$, we can calculate that

$$F = \frac{t_u \log\left(\frac{t_u}{1/2 - w_q}\right) - t_u \log(2t_u) - \alpha d(t_q \| w_q)}{d(t_u \| 1/2)} = \alpha + \frac{\alpha \log(1 - w_q) - t_u \log(1 - 2w_q)}{d(t_u \| 1/2)}.$$

Let us first consider the term $\alpha \log(1 - w_q) - t_u \log(1 - 2w_q)$. If we were to set $t_u = 1/2$, this equals

$$\left(1 + \frac{1}{\log w_q}\right) \cdot \log(1 - w_q) - \frac{1}{2}\log(1 - 2w_q) = \log(1 - w_q) - \frac{1}{2}\log(1 - 2w_q) + \frac{1}{\log w_q} \cdot \log(1 - w_q) = -\frac{w_q}{\log w_q} \pm O(w_q^2).$$

For sufficiently small $w_q$, $|\log(1 - 2w_q)| \leq 3w_q$, which means that

$$\alpha \log(1 - w_q) - t_u \log(1 - 2w_q) = -\frac{w_q}{\log w_q} \pm O(w_q^2 + |t_u - 1/2| \cdot w_q).$$

Note that $-\frac{w_q}{\log w_q}$ is positive, since $\log w_q$ is negative. If $w_q$ is sufficiently small and $|t_u - 1/2| \leq -\frac{c}{\log w_q}$ for some sufficiently small constant $c$, then the term $O(w_q^2 + |t_u - 1/2| \cdot w_q)$ is smaller than $-\frac{w_q}{\log w_q}$, which means $\alpha \log(1 - w_q) - t_u \log(1 - 2w_q) \geq 0$. Because $d(t_u \| 1/2)$ is nonnegative, this means $F \geq \alpha \geq \alpha - w_q^{1 - o(1)}$. Alternatively, if $|t_u - 1/2| \geq -\frac{c}{\log w_q}$, then we still have $\alpha \log(1 - w_q) - t_u \log(1 - 2w_q) \geq -O(w_q^2 + |t_u - 1/2| \cdot w_q) \geq -O(w_q)$, and by Proposition A.3, $d(t_u \| 1/2) = H(t_u) = \Theta((t_u - 1/2)^2) \geq \Omega\left(\left(\frac{1}{\log w_q}\right)^2\right)$. So, $F \geq \alpha - O\left(w_q \cdot (\log w_q)^2\right) \geq \alpha - w_q^{1 - o(1)}$. So, in either case, we have that $F(t_u, 0) \geq \alpha - w_q^{1 - o(1)}$, where the $o(1)$ term does not depend on $t_u$. $\qquad \square$

# B   Appendix: Omitted proofs of section 4

**Lemma B.1.** *The expected space usage of Algorithm 1 is $O(L\ell + Lk2^{-\ell} + nk)$.*

*Proof.* The algorithm has to store each of the $L$ sets $S_i$ which requires space $O(L\ell)$. Furthermore, it needs to store the sets $A_i$ which each have expected size $O(2^{-\ell}k)$. Indeed, the probability that a random subset of size $\ell$ is contained in a given $T_j$ is at most $2^{-\ell}$. Finally, the algorithm needs to store the sets $T_j$ which takes $O(nk)$ space. $\qquad\square$

**Lemma B.2.** *The expected query time of Algorithm 2 is $O(L\ell + \frac{k}{\varepsilon}(1 - \varepsilon/2)^\ell + \frac{n}{s})$*

*Proof.* First, the algorithm forms the set $Q$ which takes $O(n/s)$ time. Then, the algorithm goes over the $L$ sets $S_i$ until it finds an $i$ such that $S_i \subset Q$. This takes time $O(L\ell)$. Next, the algorithm goes through the indices $j \in A_i$. For each such $j$, it samples the set $U_j$ one element at a time checking if $U_j \subset T_j$. Let us first bound the expected size of $A_i$. We clearly have that $i^* \in A_i$. Indeed, $S_i \subset Q \subset T_{i^*}$ and $A_i$ lists all the $j$ such that $S_i \subset T_j$. Now for a $j \in [k]$ with $j \neq i^*$, the assumption that $\|p_j - p_{i^*}\|_1 > \varepsilon$, gives that $|T_j \cap T_{i^*}| \leq n(\frac{1}{2} - \frac{\varepsilon}{4})$. As the sampling of $S_1, \ldots, S_L$ and $Q$ are independent, we can view $S_i$ as a random size-$\ell$ subset of $T_{i^*}$. In particular, the probability that $S_i \subset T_j$ can be upper bounded by

$$\left(\frac{|T_j \cap T_{i^*}|}{|T_{i^*}|}\right)^\ell \leq \left(\frac{n\,(1/2 - \varepsilon/4)}{n/2}\right)^\ell = (1 - \varepsilon/2)^\ell$$

and so, the expected size of $|A_i|$ is at most $k(1 - \varepsilon/2)^\ell$. Finally, using the assumption that $\|p_{i^*} - p_j\|_1 > \varepsilon$ for $j \neq i^*$, and that $n/s \gg (\log k)/\varepsilon^2$, by a standard concentration bound, it holds for any $j \neq i^*$ that $|Q \cap T_j| \leq |Q|(1 - \varepsilon/4)$ with high probability in $k$. In particular, for $j \neq i^*$, we only expect to include $O(1/\varepsilon)$ samples in $U_j$ before observing that $U_j \subsetneq T_j$. In conclusion, the expected query time is $O(\frac{n}{s} + L\ell + \frac{k}{\varepsilon}(1 - \varepsilon/2)^\ell)$, as desired. $\qquad\square$

**Lemma B.3.** *Let $L = C\left(\frac{2}{1 - e^{-2/s}}\right)^\ell$ for a sufficiently large constant $C$. Assume that $L = k^{O(1)}$. Then Algorithm 2 returns $i^*$ with probability at least $0.99$.*

*Proof.* It is readily checked that $\ell = O(\log k) = O(\log n)$, and further, for $s > 10$, it holds that $\ell = O(\frac{\log k}{\log s}) = O(\frac{\log n}{\log s})$. We record this for later use.

Note that by standard concentration bounds, it holds with high probability in $k$ that $Q \subset T_j$ only for $j = i^*$. In fact, as in the previous proof, for all $j \neq i^*$, $|Q \cap T_j| \leq |Q|(1 - \varepsilon/4)$ with high probability in $k$. In particular, this implies that the algorithm with high probability never returns a $j \neq i^*$. Indeed, by a union bound, the probability of this happening is at most

$$k\Pr[U_j \subset T_j] \leq k(1 - \varepsilon/4)^{|U_j|} = k(1 - \varepsilon/4)^{C \log n/\varepsilon} \leq kn^{-C/4} \leq n^{-10},$$

where we used that $k$ and $n$ are polynomially related and $C$ is sufficiently large. In order for the algorithm to succeed, it therefore suffices to show that there exists an $i \in [L]$ such that $S_i \subset Q$. In that case, the algorithm will indeed return $p_{i^*}$ with high probability.

The expected size of $Q$ is

$$\mathbb{E}[|Q|] = \frac{n}{2}\left(1 - \left(1 - \frac{2}{n}\right)^{n/s}\right) \geq \frac{n}{2}\left(1 - e^{-2/s}\right)$$

and by a standard application of Azuma's inequality, it holds with high probability in $n$ that

$$|Q| = \frac{n}{2}\left(1 - e^{-2/s}\right) - O((n/s)^{0.51}). \tag{6}$$

Note that the probability that a single set $S_i$ is contained in a fixed set $Q$ is

$$\prod_{i=0}^{\ell-1}\left(\frac{|Q| - i}{n - i}\right) \geq \left(\frac{|Q| - \ell}{n}\right)^\ell. \tag{7}$$

Using the bounds on $\ell$ in the beginning of the proof, we find that $\ell \ll O((n/s)^{0.51})$. In particular, conditioning on the high probability event of Equation (6), the probability in Equation (7) is at least,

$$\left( \frac{1 - e^{-2/s}}{2} - O\left( \frac{1}{n^{0.49} s^{0.51}} \right) \right)^{\ell} \geq c \left( \frac{1 - e^{-2/s}}{2} \right)^{\ell},$$

for some constant $c > 0$. The probability that no $S_i$ is contained in $Q$ is at most

$$\left( 1 - c \left( \frac{1 - e^{-2/s}}{2} \right)^{\ell} \right)^{L}.$$

We thus choose $L = \frac{5}{c} \left( \frac{2}{1 - e^{-2/s}} \right)^{\ell}$ to ensure that this probability is at most $e^{-5} < 1/100$ and the result follows. $\qquad \square$

We can now prove our main theorem of Section 4.

**Theorem 4.2.** *Suppose $n$ and $k$ are polynomially related, $s \geq 2$, and that $s$ is such that[5] $\frac{n}{s} \geq C \frac{\log k}{\varepsilon^2}$ for a sufficiently large constant $C$. Let $\varepsilon > 0$ and $\rho_u > 0$ be given. There exists a data structure for the HUDE$(s, \varepsilon)$ problem using space $O(k^{1+\rho_u} + nk)$ and with query time $O\left( k^{1 - \frac{\varepsilon \rho_u}{2 \log(2s)}} + n/s \right)$.*

*Proof of Theorem 4.2.* Let us pick $L = k^{\rho_u}$. We further choose $\ell$ such that $L = C \left( \frac{2}{1 - e^{-2/s}} \right)^{\ell}$ for some sufficiently large constant $C$ as in Lemma B.3. In particular, $\ell \leq \rho_u \lg(k)$, so we obtain that the space bound from Lemma B.1 is $O(k^{1+\rho_u} + nk)$. On the other hand, the bound on the query time in Lemma B.2 is

$$O \left( k^{\rho_u} \log k + \frac{k}{\varepsilon} (1 - \varepsilon/2)^{\ell} + n/s \right) = O \left( k^{\rho_u} \log k + \frac{1}{\varepsilon} k^{1 + \rho_u \log(1 - \frac{\varepsilon}{2}) / \log\left( \frac{2}{1 - e^{-2/s}} \right)} + n/s \right),$$

as desired. Finally, by the choice of $L$ and $\ell$, it follows by Lemma B.3 that the algorithm returns $i^*$ with probability at least 0.99.

Note that while vastly simplified from the expression of Theorem 3.2 from [3], the expression for the query time in Theorem 4.2 is unwieldy. For a simplified version of the theorem, one can note that $\log(1 - \frac{\varepsilon}{2}) \leq -\varepsilon/2$ and for $s \geq 2$, we have that $\frac{2}{1 - e^{-2/s}} \leq 2s$, resulting in the claimed bound. $\quad \square$

*Remark B.4.* Theorem 4.2 is stated for the *promise* problem of Definition 4.1 where all distributions $p \in P$ with $p \neq p_{i^*}$ have $\|p - p_{i^*}\|_1 \geq \varepsilon$. However, even if this condition is not met, we can still guarantee to return a distribution $p_j$ such that $\|p_j - p^{i^*}\| \leq \varepsilon$ with probability 0.99 and only a slight increase in the query time. Indeed, as in the proof of Lemma B.3 as long as there exists an $S_i \subset Q$, the list $A_i$ will contain $p_{i^*}$. Moreover, as in the proof of that lemma, the algorithm will never return a $p_j$ with $\|p_j - p_{i^*}\| > \varepsilon$. Thus it will either return $p_{i^*}$ when it encounters it in the list $A_i$, or a distribution $p_j$ with $\|p_j - p_{i^*}\| \leq \varepsilon$. The only difference is that now the bound on the number of samples included in $U_j$ in Lemma B.2 becomes $O(\log n/\varepsilon)$ instead of $O(1/\varepsilon)$ with a corresponding blow up by a $\log n$ factor in the query time.

## C  Remark about experimental setting

*Remark C.1.* The prior work of [1] also tested their "FastTournament" algorithm on random half-uniform distributions. That algorithm works for the general problem and not just flat distributions, but its generality makes it is much less efficient for the special case we consider. From their experiments, in the setting where $k = 8192$, $n = 500$, and $S = 50$, the best setting of parameters achieves less than 98% accuracy using more than $400000$ operations (their notion of operation corresponds to querying the probability mass at a specific index for two distributions while our notion of operation corresponds to querying whether a specific index is in the support of a single distribution/sample). For a comparable setting of $k = 10000$, $n = 500$, $S = 50$, our algorithm uses fewer then $20000$

---

[5]The requirement $n/s \gg \log k/\varepsilon^2$ is the information theoretic lower bound for the density estimation problem.

operations, more than a $20\times$ improvement. In addition to having much better query time than the general algorithm, our algorithm has subquadratic preprocessing time of $O(kL\ell)$ while the tournament-based algorithm requires $O(k^2n)$ time which is prohibitive for the parameter settings we test. For these reasons, we restrict our comparisons to our Subset algorithm and the Elimination algorithm baseline, both tailored for the half-uniform setting.

