# OpenReview forum: "Statistical-Computational Trade-offs for Density Estimation"
_NeurIPS.cc/2024/Conference — NeurIPS 2024 poster_

### Official Review · Reviewer_b6AN · 2024-07-04

**Soundness:** 4
**Presentation:** 4
**Contribution:** 4
**Rating:** 7
**Confidence:** 3

**Summary:**

The paper provides a lower bound, and an upper bound w.r.t the hard instances, for the problem of Density Estimation. Their lower bound is based on the ‘list-of-points’ model of computation, which captures all the upper bounds from existing results. Their bounds quantify the necessary tradeoff between sampling complexity, runtime, and space complexity. Their theoretical results are also complemented by experiments showing an improvement over previous methods.

**Strengths:**

Significance: The paper considers the prominent problem of density estimation, which arises in various areas of machine learning. They prove a fundamental lower bound for the problem (and an upper bound for the hard instances), which paves the way for the construction of better algorithms for this problem (and for a deeper understanding of the fundamental problem).

Originality: the paper’s proof techniques are not entirely original, they start from lower bounds for set similarity from [3]. However, adapting these to their setting is non-trivial, and they have to deal with a complicated optimization problem for which closed-form solutions are not immediate, and their techniques for dealing with that seem unique.

Quality and Clarity: The paper is well written and clear, and the results are tight. The theoretical results are also complemented with experiments showing significant improvement over previous methods.

**Weaknesses:**

I do not know of any significant weakness to point out in the work. There is always room to make the work even more clear, especially since the bounds for the different complexities are not always intuitive, so maybe more diagrams could make it more clear but they are not necessary.

**Questions:**

A quick question about the model of computation: do you suspect this to be the only and the main model of computation that is relevant for this problem? Are there any known algorithms (even inefficient) that do not fit this model of computation? In other words, do you expect this lower bound to hold universally over all algorithms (although, of course, maybe not provably)?

**Limitations:**

authors have adequately addressed the limitations of their work.

---

> ### Author Rebuttal · Authors · 2024-08-06
>
> Thank you for the review! We address your question below.
>
> >A quick question about the model of computation: do you suspect this to be the only and the main model of computation that is relevant for this problem? Are there any known algorithms (even inefficient) that do not fit this model of computation? In other words, do you expect this lower bound to hold universally over all algorithms (although, of course, maybe not provably)?
>
> The list-of-points model of computation (this is the model in which we prove the lower bound) captures all known data-independent data structures for similarity search problems. Furthermore, it is the only model for which it is known how to prove strong lower bounds for such problems. As with any restricted model, it does not capture all possible algorithms. As mentioned in the paper, there exist data-dependent data structures (e.g., reference [5]) for approximate nearest-neighbor search (ANNS), but those techniques do not provide a benefit for random inputs such as in our lower bound construction. Since the submission of this paper, another interesting data-dependent technique has appeared for the ANNS problem. An exciting work by McCauley (reference below)  applies black-box function inversion to yield better ANNS data structures in the regime where the data structure has near-linear space, in some cases even slightly improving beyond the list-of-points lower bound for that problem. Yet, we don't see a way to use this idea to improve our upper bound for density estimation on flat distributions. It is an interesting question whether some data structure for density estimation can utilize function inversion, though as it can only improve the space at the cost of query time, this seems unsuitable in our setting. In any case, where the data-dependent techniques do apply for the ANNS problem, they do not qualitatively change the relationship between the query time and space exponents.
>
>
> Samuel McCauley. Improved Space-Efficient Approximate Nearest Neighbor Search Using Function Inversion. ESA 2024.

---

> > ### Comment · Reviewer_b6AN · 2024-08-10
> >
> > Thank you for addressing my question.

---

### Official Review · Reviewer_ZQja · 2024-07-12

**Soundness:** 3
**Presentation:** 2
**Contribution:** 2
**Rating:** 5
**Confidence:** 3

**Summary:**

This paper considers the discrete density estimation problem supported on n points. Specifically, given a set of k discrete distributions and samples from one distribution, we would like to recover the underlying distribution. The main focus of this paper is on the interplay between sample complexity and query complexity in solving this problem. The main contribution of this paper lies in establishing a trade-off between sample complexity and query complexity based on the data structure called “list-of-point-model”. The main message is that either we need a linear sample size or we need a number of queries proportional to k. The main tool used in this paper is based on the lower bound established in Ref. [3] and the technical contribution is on solving an implicit optimization problem for their problem.

**Strengths:**

The statistical and computational trade-off result for density estimation is interesting and explains why the results in the literature are nearly the best possible. This paper also gives a matching upper bound for some special cases.

**Weaknesses:**

It seems that the main hardness result the paper use is GapSS, which is from [3]. The reduction part from GapSS to URDE is new, but not sure what the novelty is there.

The writing in this paper is a little bit confusing and it would be helpful if the authors could explain what they mean by data structure in the paper. For example:
l5: what does it mean “preserving polynomial data structure”?
l38: “the data structures are subject to statistical-computational trade-offs”, it is unclear to me what it means.

**Questions:**

1.Could the authors explain where they incorporate the space information as a constraint in showing their lower bound? From Table 1, we know that if we allow superpolynomial space, then logarithmic sampling and query complexity are enough.
2.Does the query complexity used in this paper connect to the statistical query complexity? The latter is often used to establish statistical-computational trade-offs in statistical problems.
3.Does the result in this paper provide evidence for row 5 in Table 1, i.e., when s is a constant?
4. l110 in Definition 2.1 is confusing. I am wondering P is a dataset that consists data or distributions? I think the word "dataset" usually refers to samples.

**Limitations:**

This paper only considers discrete density estimation, while I think it might be more interesting to learn density in the continuous setting, such as in mixtures of Gaussians. It would be great if the authors could comment more on it.

---

> ### Author Rebuttal · Authors · 2024-08-06
>
> Thank you for your feedback. We address your comments below.
>
> >This paper only considers discrete density estimation, while I think it might be more interesting to learn density in the continuous setting, such as in mixtures of Gaussians.
>
> The main contribution of this paper is a strong lower bound for density estimation. We prove it for very simple discrete instances, which automatically implies the same lower bounds for any (possibly continuous) generalizations of such instances.
>
> From the upper bounds perspective, discrete density estimation (aka hypothesis selection) has had applications in learning mixtures of Gaussians and in robust learning of distributions (Daskalakis and Kamath [10], Diakonikolas et al [12]., Suresh et al. [18]). The basic idea is that the true distribution can be limited to a finite set of candidates, so that the density estimation can be run on those candidates.
>
> >The writing in this paper is a little bit confusing and it would be helpful if the authors could explain what they mean by data structure in the paper. For example: l5: what does it mean “preserving polynomial data structure”? l38: “the data structures are subject to statistical-computational trade-offs”, it is unclear to me what it means.
>
> We are sorry if the writing was unclear. Our lower bounds are proved in the “list-of-points” model (Definition 2.3) which formally defines the class of data-structures. Intuitively, the model allows us  to partition the set of input distributions into possibly overlapping groups. Given a query, we can search within a small number of adaptively chosen groups to find the most similar distributions to the query. Our main result is to show that in this model of computation, there is a non-trivial trade-off between the sample complexity of the algorithm and its runtime, when the algorithm is restricted to use polynomial space (referring to line 5). (We refer to it as "statistical-computational tradeoff"). As a concrete example, any data structure using space $k^{O(1)}$ and sample complexity $n/\log^{O(1)}(k)$, must have query time at least $k^{1-O(1)/\log\log k}$. In other words, even if we allow almost a linear number of samples, the running time must be close to linear.
>
> Note that prior work can solve the density estimation problem with logarithmically many samples (at the cost of $\Omega(k)$ runtime) or $o(k)$ runtime (at the cost of $\Omega(n)$ samples). Thus our main result demonstrates a non-trivial statistical-computational trade-off (referring to line 38), since it rules out algorithms that simultaneously have very small sample complexity and fast runtime.
>
> >1. Could the authors explain where they incorporate the space information as a constraint in showing their lower bound? From Table 1, we know that if we allow superpolynomial space, then logarithmic sampling and query complexity are enough.
>
> Please see the formal statement of Theorem 3.1 in the main body. The space parameter is directly linked to the query time. As stated there, allowing a large space means our query lower bound is smaller, as expected.
>
> >2. Does the query complexity used in this paper connect to the statistical query complexity? The latter is often used to establish statistical-computational trade-offs in statistical problems.
>
> Indeed the statistical query (SQ) model has been very successful in showing many statistical-computational trade-offs, but we are not aware of any such work which incorporates all of space, sample complexity, and query complexity, as we do in our work. Likewise, the list-of-points model has been extensively used to show lower bounds in similarity search problems without a statistical component.
> In our work, we use the list-of-points model to prove hardness for a statistical similarity search problem, thereby establishing an alternate method to show hardness for statistical problems beyond SQ. It is not clear if one can prove similar bounds under the SQ model since its standard formulation does not have a `space’ or ‘preprocessing’ component, which the list-of-points model does.
>
> >3. Does the result in this paper provide evidence for row 5 in Table 1, i.e., when s is a constant?
>
> In Theorem 3.1, $s$ needs to be at least a sufficiently large constant. We did not optimize this value since we are interested in the sublinear regime where $s = \omega(1)$.
>
> >4. l110 in Definition 2.1 is confusing. I am wondering P is a dataset that consists data or distributions? I think the word "dataset" usually refers to samples.
>
> Here we refer to “dataset” as the set of input distributions which we can preprocess. The samples from the unknown distribution are always referred to as samples.

---

> > ### Comment · Reviewer_ZQja · 2024-08-12
> >
> > I would like to thank the authors for the detailed response! I will keep my score.

---

### Official Review · Reviewer_prnQ · 2024-07-12

**Soundness:** 4
**Presentation:** 3
**Contribution:** 3
**Rating:** 5
**Confidence:** 3

**Summary:**

This paper considers the problem of constructing a data structure for density estimation: given a set of k distributions over [n], construct a data structure. Given samples from one of the distributions, use the data structure to identify a nearby distribution quickly. There is a three-way tradeoff between the number of samples, query time, and data structure space. The goal of this paper is to rule out any algorithm that is nearly optimal along all three axes: if you only have polynomial space, any sublinear sample complexity requires near-linear query time. [For contrast, with exponential space you can get logarithmic samples + time.]

**Strengths:**

* The lower bound is evidence against "good" data structures for this problem.
 * This sort of lower bound, involving all of sample complexity + query time + space complexity, is pretty new.
 * The lower bound is numerically quite tight, at least for "half-uniform" distributions, for o(n) samples.

**Weaknesses:**

* I'm not sure that NeurIPS is the best venue, this seems much more theoretical.
 * The lower bound is only for a particular kind of data structure.
 * The upper bound is just for the instances as used in the lower bound construction, and very synthetic.

Overall, it seems like this paper makes some reasonable progress towards understanding the complexity of data structures for this density estimation problem. But I don't really see a strong motivation for this problem, where one gets samples from one of a fixed, large-but-finite, set of possible discrete distributions and must identify which one.

The chief argument, AFAICT, is something like: there is a line of work using density estimation to learn mixtures of Gaussians, and this paper shows limits on improving the density estimation part as a black box.  But of course mixtures of Gaussians are probably way easier than the half-uniform distributions considered here.

**Questions:**

Are there other settings where one would like to solve this problem, so knowing it's hard is useful?

**Limitations:**

Fine.

---

> ### Author Rebuttal · Authors · 2024-08-06
>
> Thank you for your feedback. We address your comments below.
>
> > I'm not sure that NeurIPS is the best venue, this seems much more theoretical.
>
> We believe that NeurIPS is an appropriate venue, given that several related works (e.g., [4,7,18]) appeared in past NeurIPS conferences.
>
> >The lower bound is only for a particular kind of data structure.
>
>  As mentioned at the end of Section 2, the data structure model we are proving lower bounds for captures all known “data-independent” similarity search data structures such as locality-sensitive hashing. This has also been the model of choice for other problems in literature, such as nearest-neighbor search. Unfortunately, proving strong lower bounds for **general** data structures for virtually any computational problem in any domain is beyond the current lower bound techniques.
>
> > The upper bound is just for the instances as used in the lower bound construction, and very synthetic.
>
> Our upper bounds complement our lower bounds showing that our lower bound results are essentially tight for the setting they are proved in, therefore encapsulating the possibilities and limitations for this problem. We also note that the upper bound holds for general half-uniform distributions (not necessarily with random support).
>
> >But I don't really see a strong motivation for this problem, where one gets samples from one of a fixed, large-but-finite, set of possible discrete distributions and must identify which one.
>
> The main contribution of this paper is a strong lower bound for density estimation. We prove it for very simple discrete instances, which automatically implies the same lower bounds for any (possibly continuous) generalizations of such instances.

---

> > ### Comment · Reviewer_prnQ · 2024-08-09
> >
> > Thank you for your response.  I will keep my score.

---

### Official Review · Reviewer_yUic · 2024-07-12

**Soundness:** 3
**Presentation:** 3
**Contribution:** 3
**Rating:** 6
**Confidence:** 2

**Summary:**

The authors study the density estimation problem, i.e., given k distributions p1,...,pk over a domain [n] and a query based on samples from a query distribution q, the goal is to output a pi close to q in the 1-norm, i.e., |q-p_i|\le eps (realizable case). Note that usually k>>n.
This leads to a natural trade-off between number of samples, query time and data-structure space. Their upper bound is a modest improvement over Aamand et al. Their main result is Theorem 3.1, a lower bound for the URDE problem, based on a reduction from set similarity (Ahle and Knudson).

**Strengths:**

(strengths and weakness) The sample complexity of the density estimation problem is important practically and thus the lower bound in the paper is of interest. The authors do a good job of illustrating the reduction from the GapSS problem (Ahle and Knudson), but at the same time they only do an okay job of illustrating the novelty of their ideas. I feel the paper would benefit from a discussion of novelty in their techniques as opposed to just a survey of known bounds. Without that it's difficult for someone like me to judge their contribution.

**Weaknesses:**

see strengths above

**Questions:**

see strengths above

**Limitations:**

Theory paper, no concerns

---

> ### Author Rebuttal · Authors · 2024-08-06
>
> Thank you for your feedback. Please see our general response discussing the novelty of our techniques above.

---

### Author Rebuttal · Authors · 2024-08-06

**General response: The novelty of our techniques**

Thank you for the reviews! We provide a general comment here on the novelty of our techniques and have individual rebuttals for each reviewer.

We believe that the main conceptual contribution of this paper (specifically, the lower bound part) is in connecting a problem with a **statistical/computational tradeoff** (density estimation) to a  **purely computational** problem (similarity search, or GapSS). A priori, it is not clear that these two classes of problems are connected, given that the definition of GapSS assumes that the algorithm has access to the **full query** for free, in contrast to density estimation. In fact, we are not aware of a general black box reduction between the two problems.  Fortunately, we observed that a specific class of instances used to prove lower bounds for the GapSS can be modified to be applicable to density estimation.

Additionally, implementing this idea required a significant amount of technical development.  This is because the theorem of Ahle and Knudsen provides a lower bound for the GapSS problem via a complicated optimization problem. In particular, it does not give any explicit expression for the query time and space trade-off. One of our technical contributions is to mathematically solve this complicated optimization problem and provide an analytical expression which matches our upper bounds.

---

### Decision · Program_Chairs · 2024-09-25

**Decision:**

Accept (poster)

**Comment:**

This paper studies and derives basic results regarding data structures for hypothesis selection. I recommend acceptance.